# Reducing Work Withdrawal Behaviors When Faced with Work Obstacles: A Three-Way Interaction Model

**DOI:** 10.3390/bs13110908

**Published:** 2023-11-06

**Authors:** Jing Zhang, Di Su, Andrew P. Smith, Lei Yang

**Affiliations:** 1School of Humanity and Law, Social Governance Innovation Research Center, Henan Agricultural University, Zhengzhou 450046, China; zhangjingvip98@163.com (J.Z.); codybest@sina.cn (D.S.); 2Centre for Occupational and Health Psychology, School of Psychology, Cardiff University, Cardiff CF10 3AS, UK; 3School of Psychology, Xinxiang Medical University, Xinxiang 453003, China; 4Faculty of Psychology, Tianjin Normal University, Tianjin 300074, China

**Keywords:** cognitive flexibility, negative work rumination, obstructive stress, work control, work withdrawal behavior

## Abstract

Work withdrawal behavior is a type of negative reaction when employees face obstacles at work. Its negative impact on individuals and organizations has caught the attention of academic circles and managers. In this study, data from 596 full-time employees were collected using two timepoint measurements one month apart. The internal mechanism of the link between obstructive stress and job withdrawal behavior was analyzed, and the combined effects of work control and cognitive flexibility on the negative effects of obstructive stress were analyzed in terms of the work demand–control–personal model. The results showed that negative work rumination played a complete mediating role between obstructive stress and work withdrawal behavior, and cognitive flexibility, obstructive stress, and work control had a significant three-way interaction. The results suggest that more attention should be paid to the role of employee cognition to avoid employees’ withdrawal behavior in the face of work obstacles. In addition, when providing work resources to employees, the organization should also consider ensuring that work resources can be fully utilized to play a positive role in buffering work obstacles.

## 1. Introduction

Interpersonal conflicts, red tape, unfair treatment, and other obstructive pressures at work are relatively common phenomena. These obstructive pressures are difficult for individuals to overcome on their own, even if they try hard [1]. Out of consideration for protecting their resources, when faced with these obstacles, employees will consider responding negatively, withdrawing from their current work [2]. They exhibit negative behaviors, such as reducing working hours, arriving late and leaving early without a reason, negatively treating work content, and even thinking about leaving the job. These negative ways of coping with work obstacles can be collectively referred to as work withdrawal behavior. This refers to the intentional behavior of an employee in an attempt to avoid work, including physically withdrawing from the workplace (e.g., absence or tardiness), intentionally avoiding or leaving the organization [3,4], or psychologically disengaging from work (e.g., cyberloafing during work time) [5].

Work withdrawal behavior is already a common concern for organizations [6]. It will have cumulative negative effects on employees, such as weakening employees’ sense of career efficacy and reducing work performance [7]. Moreover, this withdrawal behavior is contagious. When some employees slack off at work, it will send negative psychological messages to others, and morale and work motivation within the organization will also decline. In turn, it will cause serious economic losses to the organization and hinder its long-term development. Existing studies have shown that work withdrawal behavior is related to various obstacles encountered by employees at work (e.g., uncivilized behavior in the workplace, abusive experience, and job insecurity) [8,9]. As for the internal mechanism of the connection between the two, existing studies have mainly focused on the emotional aspect [6,10]. However, emotion can only be one of the internal mechanisms through which stress affects behavior, and other mechanisms need to be paid attention to, such as cognitive variables. According to the persistent cognitive model of stress, work stress can continue to exert its influence after employees leave the office through work rumination [11]. According to Cropley and Zijlstra (2011) [12], work rumination includes negative affective rumination and positive problem-solving rumination. Studies have confirmed that the relationship between obstructive stress and problem-solving rumination fails to achieve a significant link with stability across time [13]. In other words, when faced with obstacles at work, employees only experience affective rumination, and affective rumination can positively predict work withdrawal behavior [14,15,16]. Based on these findings, this study analyzed the cognitive mechanism of obstructive stress affecting work withdrawal behavior; that is, it explored the mediating role of negative affective rumination.

It is not enough to identify the intrinsic mechanism by which obstructive stress affects withdrawn work behavior. More importantly, our goal is to help employees minimize withdrawal behavior in the face of difficult obstacles at work. Therefore, this study will further analyze the regulatory mechanism that alleviates the negative effects of obstructive stress. According to the work demand–control–individual model, work requirement, work control, and individual resources jointly affect individual stress perception; that is, the easing effect of work control on the negative impact of work demand only appears in certain types of individuals [17]. Based on this, this study starts with individual resources to explore ways to alleviate the impact of work obstacles on employees’ work withdrawal behavior.

### 1.1. Obstructive Stress and Work Withdrawal: The Mediating Role of Negative Work Rumination

The effect of various obstacles on work withdrawal behavior has always been the key point to which organization managers and related researchers pay attention. With the development of research, the categories of obstacles continue to expand, and a large number of studies have consistently confirmed that obstructive stress significantly positively predicts work withdrawal behavior [6,18]. However, existing research has not fully analyzed the internal mechanism of the connection between the two. The intermediary mechanism mainly focuses on employee emotion (e.g., negative emotion [10]; emotional exhaustion [6]) and attitude (e.g., job satisfaction [19]), and there is a lack of analysis of the mediation mechanism in the cognitive domain. Researchers in the field of work stress have found that the effects of work stress on individuals can continue after work through the role of persistent cognition. This persistent cognition is work rumination, which refers to the state in which some people ruminate over work-related issues and events outside of work [20]. Negative work stress, such as effort–reward imbalance, workplace incivility, and job insecurity, can cause negative work rumination [21], namely, repetitive thinking about negative experiences and experiencing negative emotions in the process [12]. Moreover, the significant relationship between obstructive stress and negative work rumination has been confirmed to be stable at the cross-sectional measurement level, 4-week interval measurement level, and daily measurement level [13]. Negative work rumination can prolong the cognitive presentation of stressors and becomes an intermediary mechanism for work stress to affect various outcome variables, such as health, well-being, and work performance [16]. Therefore, it is reasonable to speculate that negative work rumination may play a mediating role between obstructive stress and work withdrawal behavior. Empirical studies have also found that negative work rumination can significantly positively predict work withdrawal behavior [16], which leads to the prediction that:

**Hypothesis 1 (H1).** 
*Negative work rumination will play a mediating role between obstructive stress and work withdrawal behavior.*


### 1.2. Work Demand–Control–Personal Model

In dealing with the negative impact of work stress, researchers have paid attention to the important role of work resources and predicted that work resources could alleviate the negative effect of work demands on work outcomes [22]. For the interaction effect of work demands and work resources, current research mainly has three directions: additive effects, synergistic effects, and moderating effects. Existing studies have obtained more supporting evidence for the additive effects but only limited support for the other two effects, especially the moderating effect [23]. Analyzing the moderating effects is of great significance in finding ways to alleviate the negative effects of obstructive stress. In order to provide more empirical evidence for the moderating effects, this study analyzed the moderating effect of work resources on the relationship between obstructive stress and negative work rumination. According to the work demand–control model, high levels of work demand often result in employees experiencing high levels of stress. Obstructive demands are especially likely to lead to negative outcomes. The level of work control reflects the degree to which employees can freely choose work tasks, flexibly apply work strategies, and arrange work progress [24]. When employees have high work control, they can choose valuable new tasks with their own judgment and have greater autonomy in working method innovation and work process improvement. Therefore, work control resources can alleviate the stress experienced brought by negative work demands. Based on this, the second hypothesis of this study is:

**Hypothesis 2 (H2).** 
*Work control can negatively regulate the relationship between obstructive stress and negative work rumination. Specifically, when work control is at a higher level, the relationship between obstructive stress and negative rumination will be weaker.*


Existing researchers have found that only 10% of relevant studies support the buffering effect of work control resources on work stress when examining work demand control models [25]. Some scholars have suggested that this may be partly due to the individual differences in how individuals respond to their environment. Some studies have found that the hypothesized effects of the demand–control model vary by gender, culture, and personality [26,27,28]. This suggests that the significance of the interaction between work demands and work resources may only be valid under certain conditions. In predicting work stress effects, a comprehensive analysis of the moderating effects of individual resources may be more enlightening than considering situational factors alone or the main effect alone. Work resources are similar to personal resources in that they help to accomplish work goals and stimulate personal growth and development [29]. An individual who believes that they have the internal resources for the control and management of stressful situations perceives them as less stressful and responds less negatively [10]. As a consequence, personal variables related to control may prevent undesirable stress outcomes, such as counterproductive work behavior [10,30,31]. In 2012, Rubino and coworkers [17] found that the positive effect of job control was only significant in emotionally stable groups. Given this, they put forward the demand–control–individual model, arguing that work demands, work resources, and individual resources jointly affect the individual’s sense of stress. This theory has received support from other studies. Perry et al. [32] confirmed that the cushioning effect of work autonomy on work stress is only effective for emotionally stable employees. These studies suggest that employees with high levels of personal resources have greater mastery that helps them to deal more effectively with demanding conditions and, in turn, protects them from negative outcomes (i.e., exhaustion). Based on these, this study comprehensively analyzed the alleviating effects of individual resources and work resources on obstructive stress.

Cognitive flexibility refers to the ability of an individual to actively switch thoughts or behaviors to adapt to new situations [33]. Cognitive flexibility makes an individual aware that there are other options and alternatives in any situation, such as being willing to be flexible, adapting to the environment, and believing that he/she can be flexible [34]. Employees with high cognitive flexibility are typically responsive, confident, and insightful, and able to actively seek out other resources and shift perspective as circumstances change to solve novel problems [35]. Studies have shown that individuals with high levels of cognitive flexibility are better able to recover from negative events [36] and have higher life satisfaction [37]. Cognitive flexibility is thought to be a key factor in determining a person’s ability to manage and cope with stress [38]. It can mitigate the negative effects of stressful events on happiness [39]. In addition, individuals with a high level of cognitive flexibility can utilize external resources at a higher level [40,41], redeploy resources more efficiently [42], and thus achieve the desired learning effect and realize problem solving. Therefore, when faced with obstacles at work, employees with high levels of cognitive flexibility are better able to adapt and make full use of work control resources to reduce the negative impact of obstructive pressure on themselves. Based on the above, this study analyzed the combined effects of work control and individual cognitive flexibility on alleviating the negative effects of obstructive stress based on the demand–control–person model. We put forward Hypothesis 3:

**Hypothesis 3 (H3).** 
*There is a three-way interaction between obstructive stress, work control, and cognitive flexibility so that work control more strongly mitigates the relationship between obstructive stress and negative work rumination when cognitive flexibility is high compared to low.*


In summary, a diagram of the model underlying this study is shown in Figure 1.

## 2. Methods

### 2.1. Sample and Procedure

In order to reduce common method bias, two timepoint measurements were used to collect the data [43]. Control variables, obstructive stress, and work control were collected at time 1, and cognitive flexibility, negative work rumination, and work withdrawal behavior were measured one month later. Using a “snow ball sampling” methodology, online questionnaires were randomly distributed to full-time employees in various industries by research assistants, and these employees continued to forward the questionnaires. The participants were coded in the form of surname + the last four digits of the mobile phone number. Participants who completed the first measurement received one-third of the payment, and those who participate fully in both measurements receive the full payment.

Seven hundred and four valid questionnaires were obtained the first time, 616 valid questionnaires were obtained the second time, and 596 data were finally obtained after sorting the valid questionnaires that were answered in both cases. The mean age of all participants was M ± SD (36.56 ± 10.54), and the mean weekly working hours were M ± SD (44.83 ± 9.13). There were 254 males (42.60%), 342 females (57.40%). 240 (40.30%) participants with a junior college degree or below, 320 with a bachelor’s degree (53.70%), and 36 with a master’s degree (6.00%). The participants worked in a range of settings, with 93 (15.60%) in state-owned enterprises, 197 (33.10%) in private enterprises, 15 (2.40%) in transnational enterprises, 234 (39.30%) in governmental agencies or public institutions, and 57 (9.60%) in other occupational settings.

### 2.2. Measures

The measures employed in the present study have all been included in previous research in Europe, where they showed sufficient reliability and construct validity. Adopting Brislin’s (1970) back-translation procedure, we translated the original survey items into Chinese and then back-translated them into English [44]. Discrepancies were resolved through discussion between the two translators and the study investigators.

Negative work rumination: This was measured with the Affective Rumination subscale of the Work-Related Rumination Scale developed by Cropley et al. (2012) on a five-point Likert scale from 1 (strongly disagree) to 5 (strongly agree) [45]. There were a total of 5 questions. An example item is “Are you irritated by work issues when not at work?”. In the current study, the Cronbach’s alpha was 0.91 for affective rumination.

Obstructive stress: This was measured using the hindrance stress subscale of the Challenge-Hindrance Stress Scale developed by Cavanaugh et al. (2000) [46]. Five questions measured obstructive stress (e.g., “The lack of job security I have.”). The participants were asked to rate the degree to which the situation described in the item caused their stress using a five-point Likert scale from 1 (no stress) to 5 (a great deal of stress). In the current study, the Cronbach’s alpha of the hindrance stress subscale was 0.82.

Work control: This was measured with the Psychological Job Control Scale developed by Kossek et al. (2006) using a five-point Likert scale from 1 (very inaccurate) to 5 (very accurate) [47]. There are a total of 7 questions. An example item is “I have the freedom to work wherever is best for me—either at home or at work”. In the current study, the Cronbach’s alpha was 0.89 for work control.

Cognitive flexibility: This was measured with the Cognitive Flexibility Inventory developed by Dennis and Vander Wal (2010) using a seven-point Likert scale from 1 (strongly disagree) to 7 (strongly agree) [48]. This study retains the items in the original text with two measurements of factor loads above 0.5. Finally, eight items were retained to measure cognitive flexibility. An example of the items left is, “I often look at a situation from different viewpoints”. In the current study, the Cronbach’s alpha was 0.96 for cognitive flexibility.

Work withdrawal behavior: The withdrawal behavior subscale of the Counterproductive Behavior Scale compiled by Bennett and Robinson (2000) [49] and revised by Chinese scholar Zhang Yongjun (2012) [50] was adopted. A total of 5 items were scored on a five-point Likert scale from 1 (strongly disagree) to 5 (strongly agree). An example item is “Being late for work without permission”. In the current study, the Cronbach’s alpha was 0.91 for work withdrawal behavior.

Control variables: According to Zhang et al.’s research [16], in order to avoid the influence of irrelevant variables on the results, the selected control variables were gender ((1) male; (2) female), age, education level ((1) junior college degree or below; (2) bachelor’s degree; (3) master’s degree or above), job category ((1) state-owned enterprises; (2) private enterprises; (3) transnational enterprises; (4) governmental agencies or public institutions; other occupational settings). In addition, average working hours per week were controlled to make sure that the effects were not due to working many hours.

### 2.3. Assessment of Common Method Variance

The data in the present study were collected via self-administered questionnaires. Therefore, common method variance could inflate the strength of observed relationships [51]. Two methods were used to test the impact of common method variance. First, confirmatory factor analysis was performed on the data. The results showed that the fitting degree of the single-factor model (χ^2^/*df* = 20.10, CFI = 0.39, TLI = 0.34, GFI = 0.41, RMSEA = 0.18) was significantly lower than that of the five-factor model (χ^2^/*df* = 3.44, CFI = 0.92, TLI = 0.92, GFI = 0.85, RMSEA = 0.06). Then, we implemented the unmeasured latent method construction (ULMC), the results of which showed that adding one method factor (χ^2^/*df* = 2.18, CFI = 0.97, TLI = 0.96, RMSEA = 0.05) did not significantly improve the model degree of fit (∆CFI = 0.05, ∆TLI = 0.04, ∆RMSEA = 0.01). The results show that the degree of fit of the model with the common method variance latent variable is not significantly different from the original five-factor model. This also indicates that the common method variance effect in this study was minimal.

## 3. Results

### 3.1. Preliminary Analyses

Table 1 presents the means, standard deviations, and correlations. The results show that obstructive stress was positively correlated with negative rumination and work withdrawal. Negative rumination was positively correlated with work withdrawal. This provided some preliminary support for hypotheses H1 to H3.

### 3.2. Hypothesis Testing

The model was tested with the PROCESS macro (model 11) for SPSS developed by Hayes (2013). The bootstrap method was used for the hypothesis testing with 1000 iterations of bootstrapping. First, the mediating role of negative work rumination and the moderating role of work control were examined. The results show that obstructive stress was positively associated with negative work rumination (*B* = 0.23, SE = 0.04, *p* < 0.001) and was not significantly associated with work withdrawal (*B* = 0.22, SE = 0.16, *p* = 0.182). Negative work rumination was positively associated with withdrawal behavior (*B* = 1.07, SE = 0.16, *p* < 0.001). These findings suggest that negative work rumination is a complete mediator between obstructive stress and withdrawal behavior. Hypothesis 1 is supported. The ratio of the indirect effect is shown in Table 2. The interaction of work control and obstructive stress did not significantly predict negative work rumination (*B* = −0.03, SE = 0.04, *p* = 0.378). Hypothesis 2 is not supported.

Cognitive flexibility was then included in the analysis, which examined the three-way interaction of work demand–control–cognitive flexibility. The results show that the three-way interaction could significantly predict negative work rumination (*B* = −0.06, SE = 0.03, *p* < 0.05). A simple slope analysis was then carried out, and the results are shown in Figure 2. The results showed that individuals with a high level of work control matched with a high level of cognitive flexibility produced the least amount of negative work rumination when faced with a high level of work obstruction. Hypothesis 3 is supported.

Education level (*B* = 0.06, SE = 0.29, *p* = 0.85) and job type (*B* = 0.20, SE = 0.12, *p* = 0.11) did not significantly predict work withdrawal behavior. Gender (*B* = −0.78, SE = 0.33, *p* < 0.05), age (*B* = −0.05, SE = 0.02, *p* < 0.01), and average weekly working hours (*B* = −0.05, SE = 0.02, *p* < 0.01) were significantly negative in predicting work withdrawal behavior.

## 4. Discussion

This study confirms that work rumination plays an important role and is a complete mediator between obstructive stress and work withdrawal behavior. This suggests that more attention should be paid to individual cognition in future research on mediation mechanisms. According to cognitivism, the key to determining emotion is cognition. Therefore, the mediation of affective variables found in previous studies may, together with cognitive variables, play a sequential mediating the role between obstructive stress and work withdrawal behavior. Consistent with most previous studies [23], this study failed to demonstrate a significant moderating effect of work control on the negative effects of work stress. When cognitive flexibility is included, the effect of the three-way interaction is significant, showing that individuals with high cognitive flexibility match the high level of control resources and can significantly alleviate the work withdrawal behavior caused by negative work stress. This further supports the work demand–control–individual model. When exploring how to reduce the negative effects of work stress, individual resources and work resources should be combined for analysis. The level of individual resources determines whether work resources play a positive role.

The results of this study also show that when the level of control resources is low, individuals with high cognitive flexibility have more negative work rumination than individuals with low cognitive flexibility when faced with high levels of work obstacles. This may be explained by the fact that people with a high level of cognitive flexibility usually have higher confidence in their ability to cope with adversity and solve problems. However, when they are faced with the reality that they can rarely make their own decisions at work, control the pace of work, and plan various arrangements, then the reality is seriously inconsistent with their expectations. A shock from reality can send individuals into a state of ruminating over how to deal with obstacles at work. Conversely, for individuals with lower cognitive flexibility, more negative work rumination occurred under conditions of high levels of control resources (vs. low levels of control resources). Previous studies have found that when given the same level of information feedback, individuals with low cognitive flexibility cannot make full use of feedback information to achieve the same learning effects as individuals with high cognitive flexibility [41]. This indicates that individuals with low cognitive flexibility have limited utilization of objective resources, and the high level of work control resources may become a burden for them due to ineffective utilization, thus causing them to think repeatedly about work pressure.

These results suggest that the buffering effect of work resources on work stress is exerted on individuals who are able to utilize the resources. At the same time, if the individual’s ability resources need to effectively buffer the impact of work obstacles, they also have to have corresponding conditions in work (such as more work autonomy). This result further suggests that when analyzing the joint effect of work resources and individual resources, the matching effect of environment and individuals should also be considered [52].

### 4.1. Contribution

The work demand–resource–individual model explains the phenomenon of the absence of effects in previous studies, when only work resources were considered to buffer the impact of work pressure. However, since the model was proposed, it has not attracted enough attention, and it lacks sufficient analysis of the range of individual resources. This study provided more supporting evidence for the model and, for the first time, analyzed the joint effect of cognitive flexibility and work resources.

This study focused on the obstructive stress component of work stress and found that negative work rumination completely mediated the relationship between obstructive stress and work withdrawal behavior. Previous studies focused on the mediating role of emotion and attitude in the relationship between work stress and work withdrawal behavior. The results of this study suggest that cognitive factors may be more proximal mediating variables, which, together with affective factors, play a sequential mediating role between work stress and work withdrawal behavior.

The results further suggest that both work resources and individual resources are indispensable in alleviating the negative effects of work stress. Only when both resources are at a high level can the negative effects of work stress be effectively reduced. In order to alleviate the damage of work obstacles to employees, in addition to the organization providing corresponding work control resources, employees should also have the corresponding ability to realize the effective use of work resources. This provides organizations and individuals with a direction for their respective efforts.

### 4.2. Practical Implications

This study suggests that organizational managers should pay attention to the improvement of employees’ internal resources in addition to providing them with external support resources to help employees cope with work pressure. Cognitive flexibility is an individual resource that can be improved through training [33]. Organizations can carry out regular training to help employees improve their cognitive flexibility and improve their ability and confidence to cope with work pressure.

This study confirms that negative rumination about work is the key to a series of withdrawal behaviors when employees face work obstacles, so organizations should pay more attention to the psychological state of their employees. According to existing studies, mindfulness training and cognitive behavioral therapy can reduce the generation of negative work rumination [53,54,55]. Therefore, the organization could consider carrying out corresponding training for employees regularly to help them reduce negative work rumination.

### 4.3. Limitations and Future Research

Despite the unique contribution of this study, some limitations need to be addressed. First, all measures were self-reported. The time lag design helps reduce this concern, however, as did the common method variance analyses. Although the statistical results show that the common method deviation does not have much impact on the research results, to improve the reliability of the research results, the third-party evaluation method can be considered in future research. The work withdrawal behavior measured in this study was self-assessed by employees, which may also have led to a social approval effect. In the future, third-party assessments (such as leadership assessments and watching attendance clock records) should be used to obtain more objective and effective data. Secondly, according to the work demand–control–individual model, the work resource selected in this study is work control, which is a resource that has received widespread attention. However, there are splendid types of resources in the work field, so future research should pay more attention to other types of work resources to further enrich the specific details of the work demand–control–individual model.

## 5. Conclusions

This study demonstrated that the internal mechanism of the link between obstructive stress and work withdrawal behavior is negative work rumination. The dual interaction between work control and individual cognitive flexibility can alleviate negative work rumination caused by obstructive stress. The results of this study suggest that organizations should consider the matching of job resources and individual resources when providing job resources to employees to reduce the negative impact of work stress.

## Figures and Tables

**Figure 1 behavsci-13-00908-f001:**
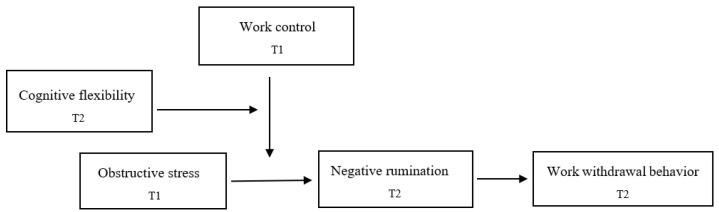
Conceptual Model. T1 = Time 1; T2 = Time 2.

**Figure 2 behavsci-13-00908-f002:**
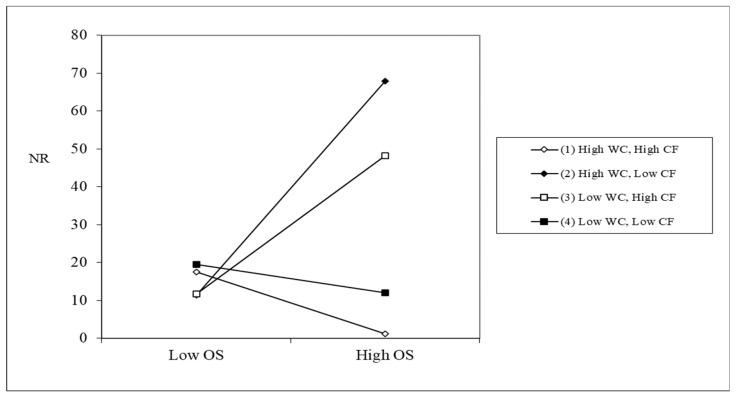
Simple slope effect diagram. *N* = 596. OS = obstructive stress; WC = work control; CF = cognitive flexibility; NR = negative rumination.

**Table 1 behavsci-13-00908-t001:** Descriptive statistics and correlations among the study variables.

	1	2	3	4	5	6	7	8	9	10
1. Work Control	1									
2. Obstructive Stress	−0.03	1								
3. Work Withdrawal	0.04	0.14 **	1							
4. Cognitive Flexibility	0.03	0.04	−0.30 **	1						
5. Negative Work Rumination	0.01	0.25 **	0.31 **	−0.04	1					
6. Age	−0.08	−0.14 **	−0.17 **	−0.07	−0.19 **	1				
7. Gender	−0.07	−0.02	−0.04	−0.09 *	0.05	−0.04	1			
8. Education	−0.04	0.11 **	0.08 *	0.13 **	0.03	−0.39 **	−0.02	1		
9. Job Category	−0.02	0.06	0.09 *	−0.15 **	0.09 *	0.11 **	0.13 **	0.05	1	
10. Weekly Working Hours	−0.02	0.03	−0.10 *	0.11 **	−0.02	−0.09 *	−0.21 **	−0.06	−0.17 **	1
*M*	21.75	12.84	9.75	39.82	12.69	36.56	1.57	1.66	2.94	44.83
*SD*	6.14	3.96	4.05	7.95	4.04	10.54	0.50	0.59	1.32	9.13

*N* = 596; * *p* < 0.05; ** *p* < 0.01.

**Table 2 behavsci-13-00908-t002:** Bootstrapping indirect effect and 95% confidence interval (CI) for the mediating effect.

PV	MV	OV	IEV	Boot SE	Bias-Corrected 95% CI	RITE
OS	NR	WW	0.29	0.07	[0.17, 0.43]	54%

*N* = 596. PV = predictive variable; MV = mediating variable; OV = outcome variable; IEV = indirect effect value; CI = confidence interval; RITE = ratio of indirect to total effect; OS = obstructive stress; NR = negative rumination; WW = work withdrawal.

## Data Availability

The data presented in this study are available upon request from the first author. The data are not publicly available because of privacy policy restrictions.

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
