# Peer review of "Reducing Work Withdrawal Behaviors When Faced with Work Obstacles: A Three-Way Interaction Model"

_behavsci, 2023, doi:10.3390/bs13110908_

Round 1

Reviewer 1 Report

Comments and Suggestions for Authors

Hello,

Thank you very much for allowing me to review the manuscript entitled Reducing Work Withdrawal Behaviors When Faced with Work Obstacles: A three-way Interaction Model.

The following is a summary of the needed revisions.

1.      Copy editing is needed as I found quite a few grammatical errors throughout the manuscript. For example, line 35 has “without reason”, which should be without a reason or without reasons; line 43 “It will not only …” does not specifically what “it” refers to; line 45 has “what’s more”, which is not a formal way to point out issues; line 182 has “network link”, which should be weblink. These are just some examples, and the authors should go through the whole manuscript and complete a copy editing.

2.      Overarching theory. The manuscript is developed based on the work-demand control model, but this model was not adequately discussed in the introduction section. Authors may want to discuss the model before the hypothesis development section.

3.      The authors should add a section about unaddressed issues, followed by a plan of study (e.g., this study will examine/test …, finding …), and a brief discussion of contributions.

4.      In line 99, hypotheses should start with a capitalized letter. This applies to the rest of the hypotheses.

5.      Hypothesis 2 should be more specific about the moderation effect. For example, … moderates the relationship between … and …, such that the relationship will be stronger/weaker when the moderator is at a higher/lower level.

6.      Figure 1 needs to be re-formatted as the current design is not well developed.

7.      Line 178 mentioned reducing common methods variance, but was not referenced. Citation should be added.

8.      In line 179, the mention of control variables is repetitive since line 231 mentioned them again.

9.      The source of sampling needs to be reported in the methods section (e.g., where was the data collected?).

10.   Line 231 mentioned all the control variables, but the authors may want to discuss why these control variables were selected.

11.   Line 237 discussed testing common method variance, however, this test is not rigorous since it is essentially a confirmatory factor analysis rather than a common method variance. The authors should consider a more rigorous approach, such as using a post-hoc method to test if common method bias is an issue in the study. For example, the unmeasured latent method constructs (ULMC).

Williams, L. J., & McGonagle, A. K. (2016). Four research designs and a comprehensive analysis strategy for investigating common method variance with self-report measures using latent variables. Journal of Business and Psychology, 31(3), 339-359.

12.   The correlation matrix needs to be re-aligned.

13.   Line 349. A theoretical implication discussion is needed. The authors can try to move part of the discussion from lines 288-327 to this part.

14.   Studies conducted. This manuscript only includes one study, with online panel data and a single-source report. This is inadequate to support these hypotheses (not to mention part of the hypotheses were not supported). The authors should consider adding another study to replicate the results in the current study. 

I hope that the above comments help you improve the quality of your manuscript.

Good luck!

Reviewer

Comments on the Quality of English Language

Copy editing is needed as I found quite a few grammatical errors throughout the manuscript. For example, line 35 has “without reason”, which should be without a reason or without reasons; line 43 “It will not only …” does not specifically what “it” refers to; line 45 has “what’s more”, which is not a formal way to point out issues; line 182 has “network link”, which should be weblink. These are just some examples, and the authors should go through the whole manuscript and complete a copy editing.

Author Response

Dear reviewer,

Thank you very much for your constructive suggestions.

My answers are attached, please check.

Reviewer 2 Report

Comments and Suggestions for Authors

 The topic is relevant to today's changing workplace environment. The writing and logical flow are sufficient. I just have a few questions / suggestions:

- The survey participants have not been grouped into different demographic categories for further more detailed analysis. I think that is needed for better understanding of the research questions.

- The survey questions are based on a 1970 study. More explanation (with references) needed on why such a historical study was used as the foundation of this new research.

Thank you!

Author Response

Dear reviewer,

Thank you very much for your constructive suggestions.

Our answers are below, please check.

Reviewer 3 Report

Comments and Suggestions for Authors

The research topic is interesting and of great academic significance and practical value, the literature review is relevant and the theoretical arguments are relatively logical, the research design is suitable, the data analysis are appropriate, and the conclusions obtained are relatively reliable. Overall, this article has the potential to contribute to the theories and is worth publishing, but there are still some issues to be improved.

1. The argument of the mediating effect of negative work rumination should be expanded and better organized. Even though previous studies have demonstrated the effects of obstructive stress on negative work rumination and the effect of negative work rumination on work withdrawal behaviors, it may help the readers to understand the mechanism by explaining the logic briefly.

2. In the argument of the moderating effect of work control, most of the spaces were spent on arguing the significance of the moderating effect, while the logic of the effect remained unexplained. Just a sentence of "According to the work demand-control model..." is not convincing enough. Please illuminate what will be the case when work control is high VS. low.

3. In the argument of the three-way interaction, the readers would expected to see how cognitive flexibility moderates the effect of work control, but most of the current arguments were still discussing how cognitive flexibility influences individuals coping with stress (line 153-line 157, cognitive flexibility moderates the effect of stress, but not a three-way interaction). So, these arguments are not enough to support the "therefore" in line 157.

4. Why was cognitive flexibility measured at time point 2? If cognitive flexibility moderates the effect of work control, shouldn't it take place and be measured before work control or at least at the same time point?

5. "Two methods were used to reduce the impact of common method variance", actually, the second method of CFA was not able to "reduce the impact but just to test whether the impact was severe.

6. Some implications seems to be not very practical. For example, even though some literature argued that cognitive flexibility can be improved through training, it may be easier to reduce obstructive stress directly.

Author Response

Dear reviewer,

Thank you very much for your constructive suggestions.

Our answers are attached, please check.

Reviewer 4 Report

Comments and Suggestions for Authors

Dear Author/s,
Firstly I would like to thank you for the opportunity to read this interesting study, which deals with an important topic. The paper is interesting, but there are some serious changes that could further improve the relevance of the study:

  1)   Theoretical background – What’s your research purpose and research question? I suggest the author should be pay more attention in Introduction part. And the literature is well-reviewed and hypotheses somehow well supported. However, I think it is necessary to add more recent studies dealing with the concept of employees’ withdrawal behaviors and work obstacles. Some suggestions:

•        Li, L., & Zheng, X. (2023). Does subordinate moqi affect employee voice? The role of work engagement and role stress. Journal of Organizational Change Management36(5), 738-754.

•        Wu, T. J., Yuan, K. S., Yen, D. C., & Yeh, C. F. (2023). The effects of JDC model on burnout and work engagement: A multiple interaction analysis. European management journal41(3), 395-403.

•       Janani, M., & Vijayalakshmi, V. (2023). An arts-based process to build Workforce agility. Journal of Organizational Change Management. doi.org/10.1108/JOCM-03-2023-0092

•         Kato, T. (2023). The mediating role of experiential avoidance in the relationship between rumination and depression. Current Psychology. doi.org/10.1007/s12144-023-05199-4

 2)     Hypotheses development: This section is poor, what is the main theory you used in this study? I thought you should to provide more detail information and convince to evident every hypothesis.

 3)     Methodology: How long is the interval between each period (time1, time2 and time 3)? How do you contact participants and who are they? I thought the author need provide more information about research procedures.

 4)     Result: Table 1 should include control variables. The MEAN are all high than 5, are you sure? I don’t think the correlation table can evident the H1-H3. The author need to do the regression analysis.

5)     The contribution seems relevant. However, there is not enough evidence for the research model developed in this study. There is not enough theoretical evidence to explain the connection among all variables in this study. It is recommended that theories or concepts be presented that may explain the relevance of the entire research model through literature review.

 6)     There are many grammars and typing error in this manuscript. There is a lot of passive voice throughout. In many cases active voice would be clearer and less wordy.

Comments on the Quality of English Language

Need to be proofreading

Author Response

(The authors gave the same response as above.)

Round 2

Reviewer 1 Report

Comments and Suggestions for Authors

The manuscript is improved, but the figure that describes the model needs to be improved. 

Comments on the Quality of English Language

Minor editing of English language required

Author Response

Dear reviewer,

1) As we said last time. For the three-way Interaction Model, Figure 1 is a very common conceptual model diagram.

This can be seen in 

[1] Chen, Y. F. ,  Crant, J. M. ,  Wang, N. ,  Kou, Y. , &  Sun, R. (2021). When there is a will there is a way: the role of proactive personality in combating covid-19. Journal of Applied Psychology, 106(2). 

[2] Jeou-Shyan, Horng, Chang-Yen, Tsai, Da-Chian, & Hu, et al. (2015). The role of perceived insider status in employee creativity: developing and testing a mediation and three-way interaction model. Asia Pacific Journal of Tourism Research.

We were not presenting a graph of statistical results, so there were no regression coefficients in Figure 1. But in order to improve the model, we marked the measurement time of each variable as T1 or T2, which was highlighted in red in the text.

2) Regarding the grammar issue, we have asked a professor from the School of Psychology at Cardiff University, a native of the UK, to check it and also run it  through Grammarly. We have not found any further modifications needed.

Could you give specific examples of modifications?

Thank you.